# A Role in 15-Deacetylcalonectrin Acetylation in the Non-Enzymatic Cyclization of an Earlier Bicyclic Intermediate in *Fusarium* Trichothecene Biosynthesis

**DOI:** 10.3390/ijms25084288

**Published:** 2024-04-12

**Authors:** Yoshiaki Koizumi, Yuichi Nakajima, Yuya Tanaka, Kosuke Matsui, Masato Sakabe, Kazuyuki Maeda, Masayuki Sato, Hiroyuki Koshino, Soichi Sato, Makoto Kimura, Naoko Takahashi-Ando

**Affiliations:** 1Graduate School of Science and Engineering, Toyo University, 2100 Kujirai, Kawagoe 350-8585, Japan; s46d02200015@toyo.jp (Y.K.); sato027@toyo.jp (S.S.); 2Graduate School of Bioagricultural Sciences, Nagoya University, Furo-cho, Chikusa-ku, Nagoya 464-8601, Japan; ynakajima.gm@gmail.com (Y.N.); painominoru24@yahoo.co.jp (Y.T.); matsuik.toyo@gmail.com (K.M.); kmaeda@agr.nagoya-u.ac.jp (K.M.); 3Faculty of Science and Engineering, Toyo University, 2100 Kujirai, Kawagoe 350-8585, Japan; sakabe@toyo.jp; 4Plant & Microbial Engineering Research Unit, Discovery Research Institute (DRI) RIKEN, 2-1 Hirosawa, Wako 351-0198, Japan; amenimomakezu_0624@major.ocn.ne.jp; 5Molecular Structure Characterization Unit, Technology Platform Division, Center for Sustainable Resource Science (CSRS) RIKEN, 2-1 Hirosawa, Wako 351-0198, Japan; koshino@riken.jp

**Keywords:** trichothecene mycotoxins, bicyclic intermediates, *Fusarium graminearum*, biosynthesis gene, second cyclization, 15-deacetylcalonectrin

## Abstract

The trichothecene biosynthesis in *Fusarium* begins with the cyclization of farnesyl pyrophosphate to trichodiene, followed by subsequent oxygenation to isotrichotriol. This initial bicyclic intermediate is further cyclized to isotrichodermol (ITDmol), a tricyclic precursor with a toxic trichothecene skeleton. Although the first cyclization and subsequent oxygenation are catalyzed by enzymes encoded by *Tri5* and *Tri4*, the second cyclization occurs non-enzymatically. Following ITDmol formation, the enzymes encoded by *Tri101*, *Tri11*, *Tri3*, and *Tri1* catalyze 3-*O*-acetylation, 15-hydroxylation, 15-*O*-acetylation, and A-ring oxygenation, respectively. In this study, we extensively analyzed the metabolites of the corresponding pathway-blocked mutants of *Fusarium graminearum*. The disruption of these *Tri* genes, except *Tri3*, led to the accumulation of tricyclic trichothecenes as the main products: ITDmol due to *Tri101* disruption; a mixture of isotrichodermin (ITD), 7-hydroxyisotrichodermin (7-HIT), and 8-hydroxyisotrichodermin (8-HIT) due to *Tri11* disruption; and a mixture of calonectrin and 3-deacetylcalonectrin due to *Tri1* disruption. However, the Δ*Fgtri3* mutant accumulated substantial amounts of bicyclic metabolites, isotrichotriol and trichotriol, in addition to tricyclic 15-deacetylcalonectrin (15-deCAL). The Δ*Fgtri5*Δ*Fgtri3* double gene disruptant transformed ITD into 7-HIT, 8-HIT, and 15-deCAL. The deletion of *FgTri3* and overexpression of *Tri6* and *Tri10* trichothecene regulatory genes did not result in the accumulation of 15-deCAL in the transgenic strain. Thus, the absence of Tri3p and/or the presence of a small amount of 15-deCAL adversely affected the non-enzymatic second cyclization and C-15 hydroxylation steps.

## 1. Introduction

*Fusarium graminearum*, a notorious plant pathogen, poses a significant threat to vital cereal crops. This fungus causes wheat head blight and maize ear rot and contaminates grains with trichothecene mycotoxins [1,2,3,4,5]. Trichothecenes, a diverse group of ribotoxic secondary metabolites, are produced by several fungal genera, including *Fusarium*, *Myrothecium, Trichothecium*, and *Trichoderma*. These sesquiterpenoid compounds share a common trichothecene skeleton (12,13-epoxytrichothec-9-ene [EPT]) in their structures, with variations in oxygenation and esterification patterns at C-3, C-4, C-7, C-8, and C-15 [6]. The trichothecene family comprises over 200 reported compounds [7]. Among the C-3 oxygenated *Fusarium* trichothecenes (previously categorized as t-type trichothecenes [6,8]), deoxynivalenol (DON) and nivalenol (NIV) produced by *F. graminearum* are classified as type B, while the T-2 and HT-2 toxins produced by *F. sporotrichioides* are classified as type A [9,10,11].

The immediate precursor of the trichothecene biosynthetic pathway is farnesyl pyrophosphate derived from the mevalonate pathway [12,13,14]. Its upstream precursor, hydroxymethylglutaryl-CoA (HMG-CoA), is synthesized by Claisen and aldol condensations of acetyl-CoA. The early biosynthetic pathway of *Fusarium* trichothecenes was established by identifying minor metabolites, (radiolabeled) precursor feeding, kinetic pulse-labeling experiments, and biosynthetic pathway-blocked mutant analyses [15]. The initial step in trichothecene biosynthesis begins with the cyclization of farnesyl pyrophosphate to the bicyclic precursor trichodiene (TDN) by an enzyme encoded by *Tri5* (originally designated as *Tox5*) [16]. Subsequently, a multifunctional cytochrome P450 monooxygenase encoded by *Tri4* [17,18] sequentially incorporates four oxygen atoms into a TDN to yield isotrichotriol (Figure 1) [19]. This bicyclic intermediate then undergoes non-enzymatic second cyclization to produce the first tricyclic trichothecene precursor, isotrichodermol (ITDmol) [20,21]. ITDmol is further modified through 3-*O*-acetylation, 15-hydroxylation, and 15-*O*-acetylation, which are catalyzed by enzymes encoded by *Tri101*, *Tri11*, and *Tri3*, respectively (Table 1). Consequently, the tricyclic precursor undergoes a sequential transformation: ITDmol → isotrichodermin (ITD) → 15-deacetylcalonectrin (15-deCAL) → calonectrin (CAL) [6,15]. Following CAL formation, the type B trichothecene biosynthesis pathway is postulated to originate from the type A pathway as a result of the coevolution of *Tri1* (A-ring hydroxylation) and *Tri13* (C-4 hydroxylation) during the diversification of trichothecene-producing fusaria [22]. The final trichothecene products are transported outside the cells by a major facilitator superfamily transporter encoded by *Tri12*. In contrast to its essential roles in T-2 toxin biosynthesis and self-protection [23], *F. graminearum* is less dependent on this efflux pump [24,25,26]. The type B trichothecene producers seem to have evolved optional toxin/drug transporter genes that complement the *Tri12* function. The expression of these pathway and transporter *Tri* genes relies on the *Tri6* and *Tri10* regulatory genes that reside in the core region of the trichothecene gene cluster [27,28,29].

In general, the blocked mutants in the secondary metabolite biosynthesis accumulate the substrate of the mutated enzyme as the primary metabolite, occasionally giving rise to shunt metabolites. Thus, the accumulation of the immediate precursor of the interrupted trichothecene pathway aids in the functional identification of the mutated Tri gene. This principle was employed in elucidating the functions of *Tri* genes involved in the biosynthetic steps, such as *FsTri11* [30], *FsTri3* [31], and *FsTri1*, using T-2 toxin-producing *F. sporotrichioides* [32]. Analyses of the major metabolites of each Δ*Fstri* mutant led to the identification of the substrates of the abovementioned three gene products: ITD, 15-deCAL, and CAL. However, a thorough examination of old literature revealed an unusual phenomenon in the production of shunt metabolites from the Δ*Fstri3* mutant, which was expected to produce 15-deCAL. Although the legitimate tricyclic precursor and its shunt metabolites were identified from all three Δ*Fstri* mutants, significant amounts of the bicyclic precursor isotrichotriol and four derived bicyclic shunts were also isolated from a UV-mutagenized Δ*Fstri3* strain of MB2972 [19]. To comprehend the potential mechanism behind bicyclic intermediate accumulation through *Tri3* disruption, we investigated whether a similar observation could be made with *F. graminearum*, which produces type B trichothecenes. In this study, we aimed to determine the fate of bicyclic and tricyclic trichothecene precursors in mutants where the C-15 acetylation step was blocked.

## 2. Results and Discussion

### 2.1. Unusually High Accumulation of Bicyclic Precursors, Isotrichotriol and Trichotriol, in the ΔFgtri3 Mutant FGD3 of F. graminearum

Previously, we investigated the accumulation of tricyclic intermediates and shunt metabolites in the cultures of Δ*Fgtri101* (Δ*Fgtri101::hph*, NBRC 113185), Δ*Fgtri11* (Δ*Fgtri11::hph*, NBRC 113181), and Δ*Fgtri1* (Δ*Fgtri1::hph*, NBRC 113176) mutants derived from the NIV chemotype strain MAFF 111233 (Table 2) [22,24]. The accumulated trichothecene metabolites resembled those found in *F. sporotrichioides* Δ*Fstri101* and Δ*Fstri11* mutants [30,33] and an *F. graminearum* Δ*Fgtri1* mutant [34] (Figure 2A). Additionally, an *FgTri3* disruption mutant (Δ*Fgtri3::hph*) of MAFF 111233, Strain FGD3 (Appendix A, Table 2), was generated in this study. The time-dependent trichothecene profiles of these four pathway *Tri* mutants were also examined (Figure 2A). The metabolites extracted from the cultures were analyzed by liquid chromatography with tandem mass spectrometry (LC-MS/MS) and subsequently visualized on thin-layer chromatography (TLC). The *R_f_* values of the spots, visualized using 4-(4-nitrobenzyl)pyridine (NBP)/tetraethylenepentamine (TEPA), were compared with those of trichothecene standards (Appendix A) to assign them to specific trichothecene metabolites.

As expected, tricyclic metabolites were detected in the ethyl acetate extracts of the mutant cultures: ITDmol from the Δ*Fgtri101* culture; a mixture of ITD, 7-hydroxyisotrichodermin (7-HIT), and 8-hydroxyisotrichodermin (8-HIT) with trace amounts of 8-ketoisotrichodermin (8-KITD), 8-acetoxyisotrichodermin (8-AITD), 3-deacetyl-7-hydroxyisotrichodermin, and 3-deacetyl-8-hydroxyisotrichodermin from the Δ*Fgtri11* culture [22]; and a mixture of CAL and 3-deacetylcalonectrin (3-deCAL) with negligible amounts of 4,15-diacetoxyscirpenol from the Δ*Fgtri1* culture [22]. In the culture of the Δ*Fgtri3* mutant FGD3, two primary spots were observed on TLC, with *R_f_* values of approximately 0.27 and 0.14, distinct from that of 15-deCAL (*R_f_* = 0.67), at earlier time points. Although the amount of 15-deCAL increased with prolonged incubation periods, the two former spots remained the main products of the FGD3 culture (Figure 2A).

To determine the structures of the metabolites that accumulated in the FGD3 mutant culture, purification was carried out through TLC, followed by NMR analysis. The two major spots with *R_f_* values of 0.32 (Spot 3; Appendix A, corresponding to an *R_f_* value of 0.27 in Figure 2A) and 0.19 (Spot 4; Appendix A, corresponding to an *R_f_* value of 0.14 in Figure 2A) were identified as bicyclic precursors, as previously reported in the literature: isotrichotriol [19] and trichotriol [37], respectively (Appendix A, Figure 2A). The presence of isotrichotriol, a bicyclic precursor, along with tricyclic 15-deCAL aligns with the findings from an *F. sporotrichioides* Δ*Fstri3* mutant, MB2972 [19]. In the silica gel scraped off the TLC plate (Spot 1; *R_f_* = 0.66), limited amounts of isotrichodiol and 7-HIT were identified around the 15-deCAL area in the NMR analysis (Appendix A). In areas with an *R_f_* value of 0.44 (Spot 2), 8-hydroxy-15-deacetylcalonectrin (8-H-15-deCAL) (Appendix A) was detected as a minor metabolite. Through LC-MS/MS analysis of the mutant culture, various tricyclic shunt products, such as 8-KITD, 8-AITD, 8-HIT, and 7-hydroxy-15-deacetylcalonectrin (7-H-15-deCAL) [22], were also identified in our in-house MS/MS library (Appendix A).

Quantification of the relative amounts of major bicyclic and tricyclic precursors, including their shunts, was performed by measuring their peak areas in the extracted ion chromatograms (XICs) obtained from the LC-MS/MS analysis (Figure 2B). When the calculated mass of *m*/*z* 286.201 corresponding to [isotrichotriol + NH_4_]^+^ and [trichotriol + NH_4_]^+^ was monitored, their possible precursor ions were also detected from the Δ*Fgtri11* culture, although the peaks were significantly smaller compared with those of the Δ*Fgtri3* culture. The same XIC peaks were marginally detected from the Δ*Fgtri101* and Δ*Fgtri1* cultures. Thus, among the sequential steps after the tricyclic ITDmol to CAL (catalyzed by Tri101p, Tri11p, Tri3p, and Tri1p, in that order), the third 15-*O*-acetylation step played a crucial role in the efficient synthesis of the tricyclic trichothecene skeleton from the bicyclic precursor. Notably, the accumulation of isotrichotriol and trichotriol in the FGD3 mutant was evident on Day 4, when 15-deCAL was barely detected (Figure 2B). This finding suggests that even a small amount of 15-deCAL effectively inhibited the second cyclization and delayed its accumulation.

### 2.2. Inefficient 15-Hydroxylation of ITD in the ΔFgtri5ΔFgtri3 Double Mutant FGD5/3 Fed with ITD

In light of the Δ*Fgtri3* mutant FGD3’s unusual accumulation of bicyclic metabolites and the limited formation of tricyclic precursors, we determined whether the formation of 15-deCAL negatively affected ITD C-15 hydroxylase activity. To address this, an ITD feeding experiment was conducted using the trichothecene-non-producing Δ*Fgtri5* mutants with additional deletions in *Tri3* (FGD5/3; Δ*Fgtri5::neo*, Δ*Fgtri3::hph*) [36] and *Tri11* (FGD5/11; Δ*Fgtri5::neo*, Δ*Fgtri11::hph*) (Appendix A). The metabolites were separated by TLC and visualized by NBP/TEPA, with *R_f_* values referencing those of FGD3 developed with the same solvent system (Appendix A). When fed to FGD5/11 (Figure 3), the ITD was converted into the resultant shunt metabolites, 7-HIT (*R_f_* = 0.65) and 8-HIT (*R_f_* = 0.53), as confirmed by LC-MS/MS (Appendix A). These transformations mirrored those observed in the shunt pathway of the Δ*Fgtri11* single mutant. During FGD5/3 feeding, the shunt metabolites 7-HIT (*R_f_* = 0.65), partially overlapping with 15-deCAL (*R_f_* = 0.67), and 8-HIT were unequivocally detected by TLC and LC-MS/MS (Figure 3), although they were barely detectable on TLC from the FGD3 mutant culture. The quantities of these shunts obtained by FGD5/3 feeding were slightly lower than those obtained by FGD5/11 feeding. These results suggest that the accumulation of 15-deCAL interferes with the ITD 15-hydroxylase activity (i.e., product inhibition), leading this precursor to enter the shunt pathway via Tri1p [22]. Alternatively, the lower level of 15-deCAL production may be caused by a functional defect in ITD C-15 hydroxylase due to the disruption of potential protein–protein interactions with Tri3p.

### 2.3. Metabolites of Trichothecene Overproducer with Disruption of FgTri3

Subsequently, we investigated whether the disruption of *FgTri3* in the trichothecene-overproducing transgenic strain led to an increase in 15-deCAL production. As one of the authors’ groups has generated a series of trichothecene-overproducing strains for studying the regulatory mechanism of trichothecene biosynthesis, YN_120 was used as an ancestor strain (Appendix A) for constructing the *Tri3*-disrupted strain YN_153 (Appendix A). In YN_120, *Tri6* and *Tri10* were connected to the promoters of the *Aspergillus nidulans* translation elongation factor gene (*TEF*) and glyceraldehyde 3-phosphate dehydrogenase gene (*GPD*), respectively, within the gene cluster of the 15-acetyldeoxynivalenol (15-ADON) producer [38]. Compared with the *Tri6* overexpressor YN_004 and the *Tri10* overexpressor YN_116, YN_120 produced a greater amount of 15-ADON in the complex medium without the toxin inducer sucrose [39] and high carbon/nitrogen ratio (Appendix A). In the Δ*Fgtri3* YN_153 strain, the trichothecene C-3 esterase *Tri8* gene [40] was additionally mutated by introducing a nonsense mutation to minimize deacetylation in the blocked trichothecene pathway (Appendix A).

In comparison with the metabolites of its *Tri3*-undisrupted parent *F. graminearum* YN_149 (Appendix A), YN_153 exhibited an atypical trichothecene profile on a liquid YG medium. YN_149 predominantly produced 3,15-diacetyldeoxynivalenol (3,15-diADON), along with smaller amounts of 15-ADON and 3-acetyldeoxynivalenol (3-ADON). In the YN_153 culture, however, the legitimate precursor 15-deCAL was not detected (Figure 4). Although a small LC-MS peak of *m*/*z* 342.191 (from 3.08 to 3.12 min), showing MS/MS spectra similar to those of [8-H-15-deCAL + NH_4_]^+^, was detected, no other tricyclic trichothecenes with available MS/MS spectra in our in-house LC-MS/MS library (including 101 trichothecenes, intermediates, and shunts) [15,18,22,36,38,39,41,42,43,44,45] were observed. Other than tricyclic trichothecenes, LC-MS analysis identified several peaks at *m*/*z* 286.201, one of which exhibited an MS/MS fragmentation pattern superimposable to that of [isotrichotriol + NH_4_]^+^; isotrichotriol was visualized as a blue spot with an *R_f_* value of 0.27 on TLC. In addition, several LC-MS peaks corresponding to [C_15_H_24_O_3_ + H]^+^ (*m*/*z* 253.180) and [C_15_H_24_O_3_ + NH_4_]^+^ (*m*/*z* 270.206) were detected on the chromatogram. Although their retention times were different from that of isotrichodiol, their MS/MS spectra were similar to those of isotrichodiol, but with some differences in the relative peak intensity ratio. Thus, the structural isomers of isotrichodiol may also be present in the YN_153 culture.

### 2.4. The Mystery of the Tri3p Acetyltransferase—Does It Play a Role in the Non-Enzymatic Second Cyclization in Trichothecene Biosynthesis?

The accumulation of bicyclic intermediates in the Δ*Fgtri3* strains, FGD3 (Section 2.1) and YN_153 (Section 2.3), strongly suggested that the deletion of *Tri3* exerted inhibitory action on the second cyclization in *F. graminearum*. This observation aligns with the results obtained in *F. sporotrichioides* [19], indicating a general role in the 15-deCAL acetylation step of the *Fusarium* trichothecene biosynthesis. Among the trichothecene pathway enzymes, FgTri4p, Tri11p, and Tri1p are cytochrome P450 monooxygenases localized to the smooth endoplasmic reticulum-derived toxisome [46,47,48,49]. Consecutive oxygenation by FgTri4p and subsequent non-enzymatic second cyclization occur in this location. Unlike other biosynthetic steps, the second cyclization of isotrichotriol to ITDmol is a non-enzymatic reaction that is not activated by Tri6p and only occurs under extremely acidic conditions (pH 2–3 or lower) in vitro [18,20,37]. The equilibrium of this chemical reaction is shifted far to the left (high concentration of reactants with a marginal amount of product) under physiological conditions. However, as explained by Le Chatelier’s principle, the position of equilibrium will shift to the right (product formation) if the concentration of the product is kept extremely low by continuous transport of the product across the toxisome membrane.

The negative impact of the newly synthesized 15-deCAL on the yield of non-enzymatic cyclization prompted us to propose a model in which the accumulation of this tricyclic intermediate in the membrane interferes with the efficient removal of ITDmol from the reaction system (see lower panel of Figure 5A; depicted by a red stop sign accompanied by an orange question mark). This interference may occur presumably by competing for or blocking up a common transport system. If this is the case, the scenario of t-type trichothecene biosynthesis will be that membrane-embedded Tri11p accepts the substrate ITD outside of the toxisome membrane and releases its product 15-deCAL in the toxisome membrane (and/or inner surface of the membrane). In the presence of the FgTri3p enzyme, 15-deCAL is efficiently transferred to its active site and serves as the substrate; the subsequent reaction product CAL is released outside of the toxisome membrane (see upper panel of Figure 5A). In the absence of FgTri3p, however, 15-deCAL released within the toxisome membrane interferes with the efficient removal of ITDmol from the reaction, resulting in the inhibition of the cyclization under physiological conditions. 

Additionally, although FgTri3p was purified from the soluble fraction of the recombinant *Escherichia coli* [42,50], it may interact with Tri11p for efficient 15-hydroxylation (Figure 5A). Indeed, we encountered an unusual property with recombinant FgTri3p; the protein (with a calculated pI of 5.12) could not be recovered during the purification process by electrostatic attraction forces using an anion exchange resin. Hence, the untagged FgTri3p protein was purified through gel permeation chromatography of the concentrated flow-through fraction of the *E. coli* crude cell extracts applied to a cation exchange chromatography column at a pH level of 6.5. This process removed the contaminating *E. coli* proteins by trapping them on a negatively charged cation exchange resin [42]. We speculate that FgTri3p is a soluble peripheral membrane protein stabilized by firmly interacting with Tri11p (Figure 5A). The significance of the arrangement of the biosynthetic machinery components in the toxisome warrants comprehensive assessment in future studies.

**Figure 5 ijms-25-04288-f005:**
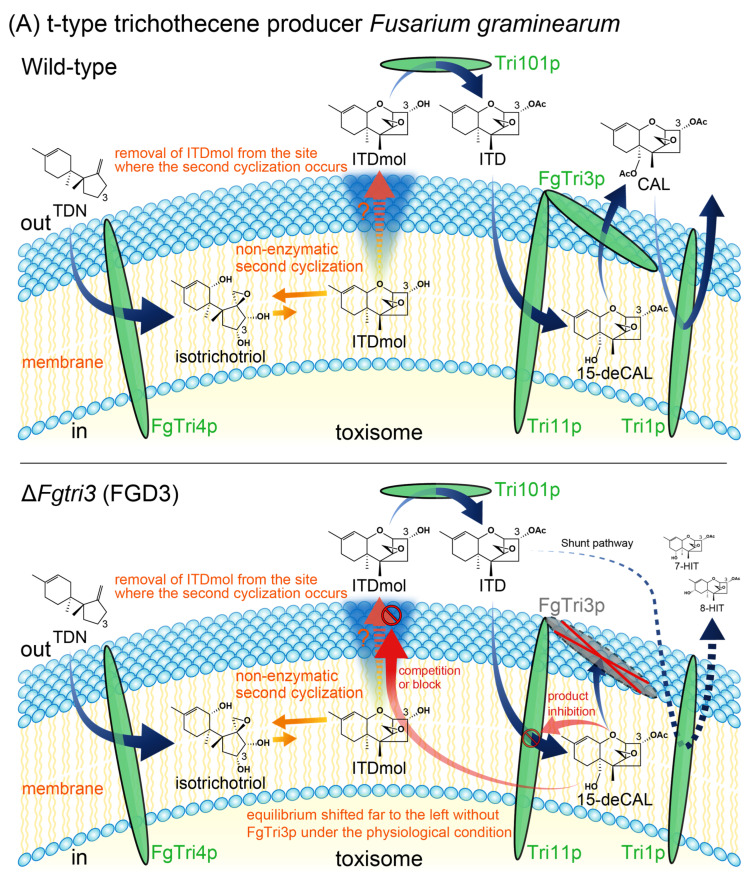
A hypothetical model elucidating the mechanism of non-enzymatic second cyclization for t-type and d-type trichothecene producers. According to Le Chatelier’s principle, the equilibrium shifts to counteract changes. The thermodynamically highly unfavorable reaction (depicted as short orange solid arrows) can be driven forward if the products were removed from the closed reaction system (depicted as orange dotted arrows). This model assumes that the Tri3p acetyltransferase is a soluble peripheral membrane protein that accepts the substrate, 15-deCAL (**A**) and trichodermol (TDmol) (**B**), from the toxisome membrane and releases the product, CAL (**A**) and trichodermin (TDmin) (**B**), outside of the toxisome. (**A**) A model elucidating accumulation of isotrichotriol and earlier bicyclic metabolites in an *F. graminearum* Δ*Fgtri3* mutant. When *FgTri3* is disrupted in the t-type trichothecene producer, 15-deCAL remains in the membrane (and/or inner surface of membrane) for an extended period, interfering with efficient removal of ITDmol from the non-enzymatic reaction system (depicted as a red stop sign accompanied by an orange question mark). The equilibrium no longer shifts forward, resulting in substantial accumulation of reactant isotrichotriol and earlier bicyclic metabolites. In this manner, the FgTri3p acetyltransferase indirectly controls the non-enzymatic second cyclization. In contrast to the Δ*Fgtri3* mutant, ITDmol, ITD, and CAL occur outside of the toxisome in the *Tri101*, *Tri11,* and *Tri1* disruption mutants, respectively. (**B**) A model elucidating accumulation of isotrichodiol and EPT in a *T. arundinaceum* Δ*Tatri3* mutant. When *TaTri3* is disrupted in the d-type trichothecene producer, reactant isotrichodiol accumulates in substantial quantities, along with EPT and TDmol [51]. The mechanism is essentially similar to t-type trichothecene biosynthesis, except that the cyclized product EPT serves as the substrate of subsequent membrane enzyme, the Tri22p C-4 hydroxylase [52] (the encoding gene previously designated as *Tri11* [53]), without prior acetylation of the tricyclic skeleton. The newly synthesized TDmol interferes with the efficient removal of EPT from the reaction system (depicted as a red stop sign accompanied by an orange question mark). Perhaps the transport of TDmol (more hydrophilic and bulkier than EPT) is less efficient than that of the first tricyclic precursor and occupies the common transport system in the membrane for an extended period. Additionally, this model suggests that both TDmin and the Tri18p acyltransferase occur outside of the toxisome. The TaTri3p acetyltransferase not only indirectly controls the non-enzymatic second cyclization, but also directly controls the localization of the Tri18p acetyltransferase’s substrate, presumably TDmol. Indeed, TDmin appears to be readily deacetylated outside of the toxisome after the enzymatic reaction by TaTri3p, as the Δ*tri18* mutant accumulates a large amount of TDmol in the medium [54]. Thus, Tri18p catalyzes a nucleophilic acyl substitution reaction between TDmol and an acyl compound 2,4,6-trienedioyl-CoA (or 2,4,6-trienedioyl-acyl carrier protein thioester), where the nucleophile (deprotonated TDmol; see inset of upper panel) displaces the leaving group, -S-CoA (or acyl carrier protein), on the electrophilic carbonyl carbon of acyl-CoA. Abbreviation: HA, harzianum A.

### 2.5. Possible Role of Tri3p in d-Type Trichothecene Producer

The role of Tri3p acetyltransferase in the second cyclization also seems to apply to the biosynthesis of harzianum A (HA) in *Trichoderma arundinaceum*. HA belongs to biosynthetically and evolutionarily distinct d-type trichothecenes that lack a C-3 hydroxy group, which is crucial for self-protection by *Tri101* (Figure 5A) [6,8,45]. Among the *Tri* genes of d-type and t-type trichothecene producers, functional changes occurred in *Tri4* and *Tri3*, resulting in the structural diversity of this important group of mycotoxins [51,53]. A similar phenomenon was observed in *T. arundinaceum* by Proctor and co-workers; that is, the disruption of *T. arundinaceum Tri3* (*TaTri3*), encoding a trichodermol (TDmol) C-4 acetyltransferase, led to the accumulation of a large amount of bicyclic isotrichodiol, along with EPT and the legitimate precursor TDmol [51]. Given that a C-4 hydroxylase encoded by *Tri22* (previously designated as *Tri11* [53]) accepts substrate EPT outside of the toxisome and releases the product TDmol in the toxisome, the absence of the TaTri3p acetyltransferase causes TDmol accumulation at the site of the second cyclization, resulting in competition with EPT for transport across the toxisome membrane (see lower panel of Figure 5B; depicted by a red stop sign accompanied by an orange question mark). Thus, the accumulation of the bicyclic (i.e., isotrichodiol) and earlier tricyclic (i.e., EPT) metabolites in the Δ*Tatri3* mutant can be explained by Le Chatelier’s principle in purely chemical reaction and product inhibition of Tri22p, respectively.

Although Δ*Tatri3* carries an intact copy of *Tri18* encoding a 2,4,6-trienedioyl transferase responsible for esterification at C-4, the mutant produces only 6−8% of wild-type levels of HA [54]. TDmol, EPT, and isotrichodiol were the major products of the mutant [51]. Assuming that the TaTri3p acetyltransferase moves TDmol outside of the toxisome as the acetylated product trichodermin (TDmin), the Δ*Tatri3* mutant’s inability to produce a wild-type level of HA may be due to the reduced metabolic flow of TDN to the C-4 oxygenated TDmol (i.e., TDN-derived carbon also distributed to isotrichodiol and EPT [51,55] and differences in the intracellular localization of this possible substrate of Tri18p. Thus, the most important role of TaTri3p in HA biosynthesis involves enabling Tri18p acyltransferase access to the C-4 oxygenated substrate, presumably the TaTri3p’s deacetylated product of TDmol. Indeed, the Δ*tri18* mutant produces a high level of TDmol in the culture medium [54], suggesting that a relatively labile C-4 acetyl group of TDmin is hydrolyzed outside of the toxisome. Determining whether the Δ*Tatri3* mutant can convert exogenously added TDmol to HA and/or the recombinant Tri18p enzyme can efficiently catalyze the esterification of TDmol using 2,4,6-trienedioyl-CoA as a co-substrate (Figure 5B) is of particular interest.

### 2.6. Acetyltransferases as Possible Targets in the Development of Agricultural Chemicals That Suppress Trichothecene Production

Trichothecenes serve as virulence factors in plant pathogenesis [56]. Indeed, *F. graminearum* mutants generated through the transformation-mediated disruption of *Tri5* are unable to produce DON and exhibit decreased virulence in wheat head blight [57,58] and maize ear rot [59], offering a novel strategy to control the plant diseases by eliminating its toxicity. Agricultural chemicals that limit trichothecene biosynthesis by the fungus during infection are expected to reduce not only mycotoxin accumulation but also the disease severity. Although not practically applicable in the field, several small-molecule compounds are recognized for inhibiting trichothecene production by repressing *Tri* gene expression under specific culture conditions. These compounds include natural phenolic acids [60,61,62,63], 2,4-dihydroxy-7-methoxy-2*H*-1,4-benzoxazin-3(4*H*)-one (DIMBOA) [64], and dihydroartemisinin [65]. Additionally, other known inhibitors target biosynthetic enzymes, such as Tri5p TDN synthase [66,67,68], Tri4p multifunctional oxygenase [41,69,70,71,72,73], and Tri101p acetyltransferase [74,75], which are important for the biosynthesis of fully toxic trichothecenes. The fact that deletion of the *Tri3* acetyltransferase gene limits the formation of toxic tricyclic ring provides an additional option for selecting biosynthetic enzymes in target-oriented inhibitor screening [76] and broadens the possibility of developing useful agrochemicals for the control of Fusarium head blight.

## 3. Materials and Methods

### 3.1. Strains

Single-gene deletion strains derived from MAFF 111233 (NIV chemotype strain of *F. asiaticum* belonging to the *F. graminearum* species complex), such as *Fgtri101^−^* #2s1* (Δ*Fgtri101::hph*, NBRC 113185) [24], *Fgtri11^−^* #2s1 (Δ*Fgtri11::hph*, NBRC 113181) [35], and *Fgtri1^−^* #1 (Δ*Fgtri1::hph*, NBRC 113176) [22], were used to analyze time-dependent trichothecene profiles (Table 2). The Δ*Fgtri5*Δ*Fgtri3* strain of FGD5/3 (Δ*Fgtri5::neo*, Δ*Fgtri3::hph*, NBRC 114122) [36] was used in ITD feeding experiments. The strains are available from the Biological Resource Center of NITE (NBRC) (Kisarazu, Japan). The Δ*Fgtri3* strain of FGD3 (Δ*Fgtri3::hph*: Appendix A) and Δ*Fgtri5*Δ*Fgtri11* strain of FGD5/11 (Δ*Fgtri5::neo*, Δ*Fgtri11::hph*: Appendix A) were generated from the wild-type (WT) and FGD5 (Δ*Fgtri5::neo*) [18] strains, respectively (Table 2). Trichothecene overproducer-derived strains are descendants of the 15-ADON chemotype of strain JCM 9873 [77]: YN_149 (Δ*Tri6*/P*_TEF_::Tri6*, ΔP*_Tri10_*/P*_GPD_*, Δ*Tri8*/*Tri8_nsm*) and YN_153 (Δ*Tri6*/P*_TEF_::Tri6*, ΔP*_Tri10_*/P*_GPD_*, Δ*Tri8*/*Tri8_nsm*, Δ*Tri3*/P*_GPD_::hph::tk*). These strains were generated from the strain YN_001 [38] through a series of transformations performed in this study (Appendix A).

### 3.2. Reagents

Hygromycin B and geneticin disulfate were obtained from Nacalai Tesque, Inc. (Kyoto, Japan) and used for maintaining mutant strains. TLC plates (silica gel 60 F_254_) were purchased from Merck KGaA (Darmstadt, Germany), while hexane, toluene, ethyl acetate, chloroform, carbon tetrachloride, and NBP/TEPA were purchased from Kanto Chemical Co., Inc. (Tokyo, Japan). The LC-MS grade acetonitrile and other reagents were purchased from Merck KGaA (Darmstadt, Germany).

### 3.3. Medium and Culture Conditions

V8 juice agar (20% Campbell’s V8 juice [*v*/*v*], 0.3% CaCO_3_ [*w*/*v*], and 2% agar [*w*/*v*]) containing appropriate antibiotics (300 µg/mL hygromycin B and/or 100 µg/mL G418), if necessary, served as the maintenance medium for fungal strains. The composition of RF and YG media followed the previously described formulations [35,64]. The YS medium comprised 0.5% (*w*/*v*) yeast extract and 2% (*w*/*v*) sucrose. The rice medium was prepared by soaking 100 g of rice in 50 mL of water and autoclaving it for 15 min. Trichothecene production in *Tri* gene disruptants derived from MAFF 111233 was induced using the RF medium for mycelial plug inoculation. In the feeding experiment involving ITD, the induced mycelia of FGD5/3 and FGD5/11 were suspended in sterilized water (300 mg of wet mycelia/mL), fed with 10 µg/mL of ITD, and cultured with gyratory shaking in a 100-mL Erlenmeyer flask. Trichothecene production of YN_149 and YN_153 (transformants derived from a 15-ADON producer JCM 9873) was induced by inoculating germinating spores (16 h after inoculation into the YG medium) into the YG medium and cultured for 5 days with gyratory shaking at 135 rpm and 25 °C. 

### 3.4. Preparation of Trichothecene Standards

The trichothecene intermediates were prepared from *FgTri* gene disruptants derived from MAFF 111233: ITDmol from Δ*Fgtri101*; ITD, 7-HIT, and 8-HIT from Δ*Fgtri11*; and CAL from Δ*Fgtri1* [22,45]. The hyphae of each disruptant were inoculated into an RF medium and incubated with gyratory shaking at 22 °C for 7 days, followed by the extraction of each trichothecene using ethyl acetate. To generate 3-deCAL and 15-deCAL, purified CAL underwent nonspecific alkaline deacetylation to produce 3,15-dideacetylcalonectrin and subsequent specific *O*-acetylations with Tri3p and Tri101p, respectively, as previously described [22]. Meanwhile, 15-ADON and DON were prepared from JCM 9873, while 3-ADON was prepared from *F. graminearum* DSM 4528 [78]. Each strain was incubated in YS medium with gyratory shaking at 22 °C for 3 days. The hyphae of each strain were inoculated into the rice medium and incubated for 1 week at room temperature. Each trichothecene was extracted in acetonitrile. The trichothecenes and their intermediates were purified using Purif-Rp2 (Shoko Science Co., Ltd., Kanagawa, Japan) equipped with Purif-Pack SI 25 (Shoko Scientific, Kanagawa, Japan). If necessary, the samples were further purified using preparative HPLC (LC-4000 series, JASCO, Corp., Tokyo, Japan; UV detection at 254 or 195 nm) equipped with a C_18_ column (Pegasil ODS SP100 10φ × 250 mm; Senshu Scientific Co., Ltd., Tokyo, Japan) [45]. 

### 3.5. Trichothecene Analyses

For the trichothecene analysis of the mutant cultures, each ethyl acetate extract was evaporated at 60 °C on a heating block and subsequently reconstituted in ethanol. Each sample was separately applied on TLC plates (silica gel 60 F_254_) using ethyl acetate/toluene (3:1) as the developing solvent. The trichothecenes were visualized using the NBP/TEPA method [79]. Each ethyl acetate extract was diluted in ethanol and filtered through a syringe filter, and 2 µL of the resulting filtrate was applied to LC-MS/MS for straightforward identification. The LC-MS/MS system comprised an Eksigent ekspert™ ultraLC 100-XL (Dublin, CA, USA) and an AB SCIEX Triple TOF 4600 System (Framingham, MA, USA), with a DuoSpray source operated in electrospray ionization mode. The LC and MS analyses were performed following the procedures outlined in our previous studies [45]. Each pattern of the MS/MS spectra was obtained in positive ion mode using information-dependent acquisition (IDA) and/or time-of-flight (TOF)-MS methods. Data were analyzed using SCIEX OS-Q2.1 (AB SCIEX).

FGD3 metabolites were purified from the ethyl acetate extract of the 1 L culture through preparative TLC. Following the identification of the zone containing trichothecene intermediates, characterized by a blue color development upon reaction with NBP/TEPA, the metabolites in the silica gel layer of the intact TLC plates were carefully scraped off using a spatula and subsequently eluted with ethyl acetate. The structures of the trichothecene intermediates were determined from ^1^H NMR and ^13^C spectra in CDCl_3_ acquired with a JEOL JNM-ECA600 spectrometer using residual CHCl_3_ (δ^1^H: 7.26) and CDCl_3_ (δ^13^C: 77.0) as an internal standard.

## 4. Conclusions

The disruption of *FgTri3*, responsible for encoding 15-deCAL 15-acetyltransferase, results in an inefficient second cyclization and the unusual accumulation of bicyclic intermediates and shunts in *F. graminearum*. This phenomenon is consistent with the findings of the Δ*Fstri3* mutant of *F. sporotrichioides* strain MB2972 [19]. The uniqueness of this phenomenon prompts further exploration by examining the localization of trichothecene pathway enzymes, their substrates, and their products in the toxisome.

## Figures and Tables

**Figure 1 ijms-25-04288-f001:**
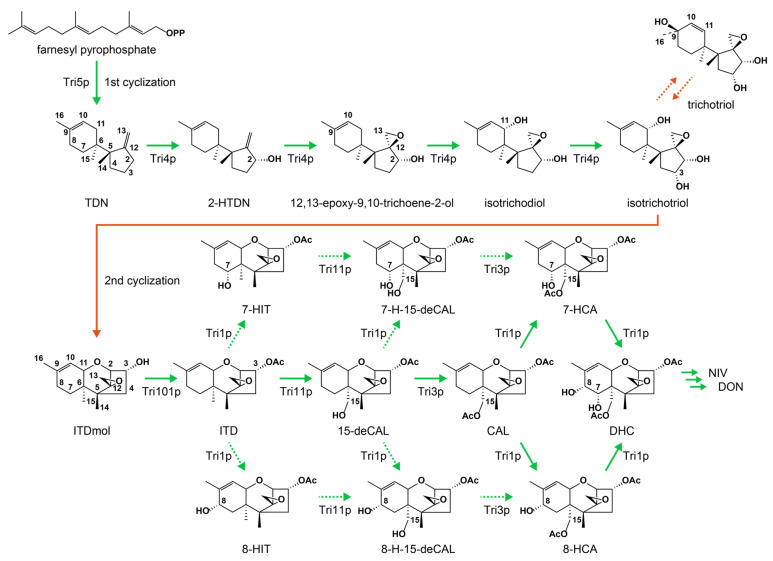
Early biosynthetic pathway of type B trichothecenes in *Fusarium* species. Green arrows represent the enzymatic reactions of the Tri enzymes, while orange arrows denote non-enzymatic reactions. Minor routes are indicated in dotted green arrows. Interconversion between two bicyclic metabolites is shown by dotted orange arrows. The *Tri3* disruptant accumulated higher levels of bicyclic metabolites, isotrichotriol and trichotriol, compared with those of the expected precursor, 15-deCAL. Abbreviations: TDN, trichodiene; 2-HTDN, 2-hydroxytrichodiene; ITDmol, isotrichodermol; ITD, isotrichodermin; 15-deCAL, 15-deacetylcalonectrin; CAL, calonectrin; DHC, 7,8-dihydroxycalonectrin; NIV, nivalenol; DON, deoxynivalenol; 7-HIT, 7-hydroxyisotrichodermin; 7-H-15-deCAL, 7-hydroxy-15-deacetylcalonectrin; 7-HCA, 7-hydroxycalonectrin; 8-HIT, 8-hydroxyisotrichodermin; 8-H-15-deCAL, 8-hydroxy-15-deacetylcalonectrin; 8-HCA, 8-hydroxycalonectrin.

**Figure 2 ijms-25-04288-f002:**
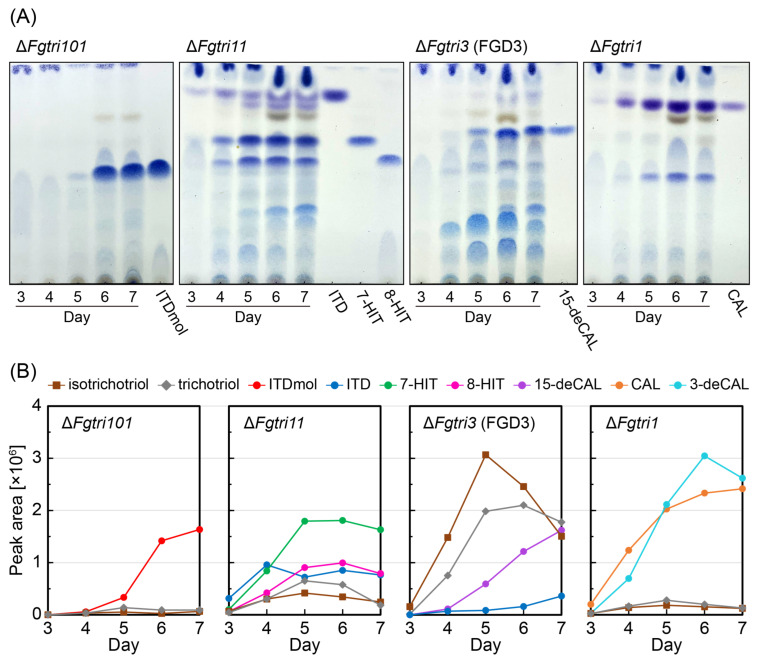
Time-dependent accumulation of trichothecenes in the culture of each *Tri* gene disruptant. (**A**) TLC analysis of trichothecenes in the cultures of Δ*Fgtri101*, Δ*FgTri11*, Δ*Fgtri3* (FGD3), and Δ*Fgtri1* mutants. One milliliter medium from each culture was sampled on Days 3, 4, 5, 6, and 7 and extracted twice with ethyl acetate. The condensate was dissolved in 10 µL of ethanol. The dissolved solution was subjected to TLC. Trichothecenes were visualized by NBP/TEPA after development using ethyl acetate/toluene (3:1) as the solvent. (**B**) LC-MS analysis of trichothecenes in the cultures of Δ*Fgtri101*, Δ*FgTri11*, Δ*Fgtri3* (FGD3), and Δ*Fgtri1* mutants. Each condensed extract, prepared as described in the legend of panel (**A**), was dissolved in 350 µL ethanol. After removing the debris using a syringe filter, 2 µL of the filtrate was applied to LC-MS. The assignment of each MS peak was confirmed by comparing its MS/MS spectra to that of the standard (Appendix A). The areas of trichothecenes’ parent MS peaks, acquired in a time-of-flight (TOF)-MS mode, were plotted for each day: isotrichotriol, [C_15_H_24_O_4_ + NH_4_]^+^ (*m*/*z* 286.201), RT 3.45 min; trichotriol, [C_15_H_24_O_4_ + NH_4_]^+^ (*m*/*z* 286.201), RT 3.04 min; ITDmol, [C_15_H_22_O_3_ + H]^+^ (*m*/*z* 251.164), RT 3.88 min; ITD, [C_17_H_24_O_4_ + H]^+^ (*m*/*z* 293.175), RT 4.95 min; 7-HIT, [C_17_H_24_O_5_ + NH_4_]^+^ (*m*/*z* 326.196), RT 3.84 min; 8-HIT, [C_17_H_24_O_5_ + NH_4_]^+^ (*m*/*z* 326.196), RT 3.44 min; 15-deCAL, [C_17_H_24_O_5_ + NH_4_]^+^ (*m*/*z* 326.196), RT 3.54 min; 3-deacetylcalonectrin (3-deCAL), [C_17_H_24_O_5_ + NH_4_]^+^ (*m*/*z* 326.196), RT 3.58 min; and CAL [C_19_H_26_O_6_ + NH_4_]^+^ (*m*/*z* 368.207), RT 4.41 min.

**Figure 3 ijms-25-04288-f003:**
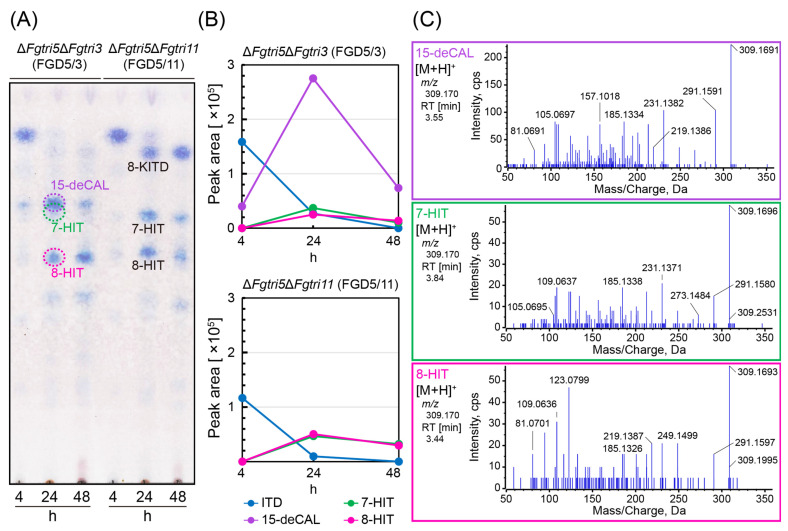
Trichothecene profiles of double gene disruptants fed with ITD. (**A**) Time-course metabolism of ITD fed to FGD5/3 and FGD5/11. After 4, 24, and 48 h of incubation, 4 mL aliquots of the medium were sampled, extracted, and condensed as described in the legend of Figure 2. Approximately 90% of each condensed sample was applied to TLC, and trichothecenes were visualized by NBP/TEPA. (**B**) Time course of parental MS peak area of ITD, 15-deCAL, 7-HIT, and 8-HIT. The rest (=10%) of the condensed samples in (**A**) were dissolved in 1 mL ethanol, and 2 µL aliquots were applied to LC-MS after filtration. Each trichothecene was measured in information-dependent acquisition (IDA) mode: ITD, [C_17_H_24_O_4_ + H]^+^ (*m*/*z* 293.175); 15-deCAL, [C_17_H_24_O_5_ + H]^+^ (*m*/*z* 309.170); 7-HIT, [C_17_H_24_O_5_ + H]^+^ (*m*/*z* 309.170); and 8-HIT, [C_17_H_24_O_5_ + H]^+^ (*m*/*z* 309.170). (**C**) The MS/MS spectra of 15-deCAL, 7-HIT, and 8-HIT transformed from ITD in this feeding assay. These MS/MS spectra were obtained from the FGD5/3 culture after 24 h of ITD feeding. They were identical to those of the standard samples (Appendix A).

**Figure 4 ijms-25-04288-f004:**
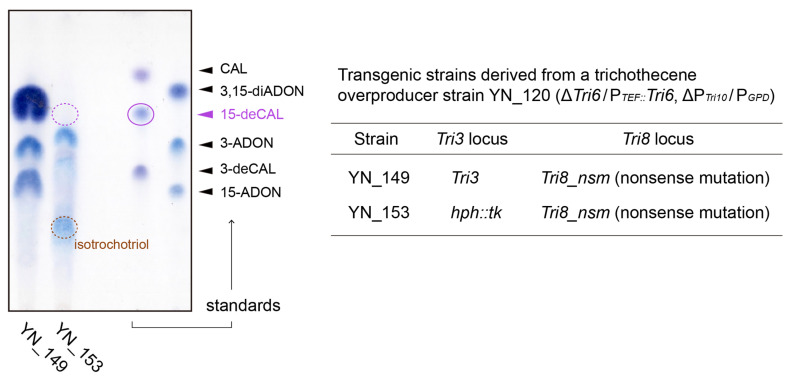
Trichothecene profiles of the toxin overproducers YN_149 and YN_153, derived from JCM 9873 (Appendix A). In YN_149, *Tri6* and *Tri10* were overexpressed, and *Tri8* was inactivated with a nonsense mutation; a functional *Tri3* gene of YN_149 was replaced by an *hph::tk* cassette in YN_153. After 5 days of incubation on a liquid YG medium with gyratory shaking (135 rpm) at 25 °C, the ethyl acetate extracts from 0.5 mL of culture were analyzed on TLC.

**Table 1 ijms-25-04288-t001:** Functions of trichothecene biosynthetic (*Tri*) genes of t-type and d-type trichothecene producers.

t-Type Trichothecene Producer (*F. graminearum*)		Function	d-Type Trichothecene Producer (*T. arundinaceum*)	
Position of the Reaction Site	Gene		Gene	Position of the Reaction Site
C-5 and C-12, C-6 and C-11 (1st cyclization)	*Tri5*	Cyclization	*Tri5*	C-5 and C-12, C-6 and C-11 (1st cyclization)
C-2, C-3, C-11, C-13	*FgTri4*	Oxygenation	*TaTri4*	C-2, C-11, C-13
C-3	*Tri101*	Acetylation	−	−
C-15	*Tri11*	Oxygenation	*Tri22*	C-4
C-15	*FgTri3*	Acetylation	*TaTri3*	C-4
C-7, C-8	*Tri1*	Oxygenation	−	−

Trichothecenes may be evolutionarily divided into two groups, t-type and d-type, respectively, based on the presence and absence of an oxygen at C-3 [6,8]. The *Tri* genes of *F. graminearum* and *T. arundinaceum* are listed from top to bottom by order of the biosynthetic pathway. *Tri* genes with different functions among trichothecene-producing species are prefixed with the abbreviated species names of *F. graminearum* (*Fg*) and *Trichoderma arundinaceum* (*Ta*). Note that the carbon number refers to the numbering system of the EPT skeleton; in the first cyclization catalyzed by Tri5p, C-5, C-6, C-11, and C-12 correspond to C-7, C-6, C-1, and C-11, respectively, of (2*E*,6*E*)-farnesyl pyrophosphate. The non-enzymatic second cyclization occurs following the oxygenation of the bicyclic precursor by Tri4p.

**Table 2 ijms-25-04288-t002:** Transgenic *Fusarium* strains in which the *Tri* genes were disrupted by double crossover homologous recombination.

Strains	Parent Strains	Genotypes	Major Trichothecene Metabolites	References
MAFF 111233	–	Wild-type (WT)	4-acetylnivalenol, 4,15-diacetylnivalenol	[35]
NBRC 113185 (*Fgtri101^−^* #2s1*)	MAFF 111233	Δ*Fgtri101::hph*	ITDmol	[24]
NBRC 113181 (*Fgtri11^−^* #2s1)	MAFF 111233	Δ*Fgtri11::hph*	ITD, 7-HIT, 8-HIT	[35]
FGD3	MAFF 111233	Δ*Fgtri3::hph*	15-deCAL, isotrichotriol, trichotriol	This study
NBRC 113176 (*Fgtri1^−^* #1)	MAFF 111233	Δ*Fgtri1::hph*	CAL, 3-deCAL	[22]
NBRC 113175 (FGD5)	MAFF 111233	Δ*Fgtri5::neo*	–	[18]
NBRC 114122 (FGD5/3)	NBRC 113175	Δ*Fgtri5::neo*, Δ*Fgtri3::hph*	–	[36]
FGD5/11	NBRC 113175	Δ*Fgtri5::neo*, Δ*Fgtri11::hph*	–	This study

The transgenic strains reported in this table, FGD3 and FGD5/11, will be publicly available from NBRC (NITE Biological Resource Center), Kisarazu, Japan. The bicyclic metabolites are underlined.

## Data Availability

Data are contained within the article and Appendix A.

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
