# Peer review of "A Role in 15-Deacetylcalonectrin Acetylation in the Non-Enzymatic Cyclization of an Earlier Bicyclic Intermediate in Fusarium Trichothecene Biosynthesis"

_ijms, 2024, doi:10.3390/ijms25084288_

Round 1
Reviewer 1 Report (New Reviewer)
Comments and Suggestions for Authors
The authors present a study focusing on the biosynthesis of trichothecene metabolites and related Tri genes. The findings are valuable for a better understanding of how Fusarium produces trichothecene mycotoxins from a genetic perspective. Using various analytical tools and molecular biological manipulations the authors have accumulated an overwhelming amount of data to identify the role of tri genes (e.g., FgTri3) and corresponding production of biomarkers (trichothecene metabolites). Below please find my comments for consideration.
1. Revise the abstract. The current version does not give a clear picture regarding the key findings of the study. Not sure which part is background and which part is the key findings.
2. Include a flowchart to illustrate the hypothesis and the study design, helping readers go through the results and the discussions.
3. Figure 3 and Supplementary Figure S3. Be consistent with decimals regarding m/z values. For example, (A) ITDmol's m/z is reported as 251.164, while in the spectrum the m/z values are reported to 4 decimal places. Furthermore, is 251.164 the theoretical value? The corresponding mass error should be calculated for the experimental value of the precursor ion.
Comments on the Quality of English LanguageMinor editing of English language required.
Author Response
Thank you very much for your helpful suggestions.
Following your suggestions, we revised our manuscript, and highlighted the modified part in yellow. When we resubmitted our revised manuscript, we also sent the same version to a professional language editing service simultaneously. Following the English polisher’s suggestions, we corrected our manuscript and highlighted our changes in gray. To improve our manuscript, we also made several more changes, which are highlighted in blue. Please check our manuscript highlighted in three colors.
I answered below to all the comments you pointed out.
- Revise the abstract. The current version does not give a clear picture regarding the key findings of the study. Not sure which part is background and which part is the key findings.
ïƒ In our abstract, we use “present tense” for background and “past tense” for our new findings of this study. In order to clarify the difference between them, we also added one more sentence in Line 23-25, “In this study, we extensively analyzed the metabolites of the corresponding pathway-blocked mutants of Fusarium graminearum.”
- Include a flowchart to illustrate the hypothesis and the study design, helping readers go through the results and the discussions.
ïƒ We modified our graphical abstract so that it might illustrate our hypothesis better. Although it is not a flowchart, it can explain our hypotheses step-by-step by each sentence.
We also divided both Figure 5A and 5B into two panels, upper and lower, which illustrate localizations of trichothecene intermediates of wild-type and ΔTri3 strain, respectively. We found our revised Figure 5 much easier for readers to understand. We appreciate the reviewer’s helpful suggestion.
- Figure 3 and Supplementary Figure S3. Be consistent with decimals regarding m/z values. For example, (A) ITDmol's m/z is reported as 251.164, while in the spectrum the m/z values are reported to 4 decimal places. Furthermore, is 251.164 the theoretical value? The corresponding mass error should be calculated for the experimental value of the precursor ion.
ïƒ All the values representing 3 decimal places are theoretical ones, while those of 4 decimal places are analytical ones. The number at 4 decimal places varies in each analysis, and in even consecutive ones. In all the MS/MS spectra, the instrument shows the values of 4 decimal places, but the number at 4 decimal places is not accurate, and does not mean much to us. The purpose of our paper is identification of trichothecene metabolites that are in our in-house MS/MS library. We have characterized them in detail in our studies for the past 11 years and are very familiar with them.

Reviewer 2 Report (New Reviewer)
Comments and Suggestions for Authors
I enjoyed reading this article that represents a panoramic study of the biosynthesis of trichothecene in Fusarium. Overall quality is impressive. The research flow is clear and experimental approach is suitable.
Few comments:
1-Please include 1-2 phrases about upstream precursors such as FPP synthesis and transport in Fusarium.
2-Include a sentence about byproducts or competing pathways.
3-Highlight better the importance of trichothecene synthesis in the fungus.
4-Check the minor comments in the pdf file.

Author Response
Please see the attachment.

This manuscript is a resubmission of an earlier submission. The following is a list of the peer review reports and author responses from that submission.
Round 1
Reviewer 1 Report
Comments and Suggestions for Authors
As a reviewer, I appreciate the comprehensive exploration of the trichothecene biosynthesis pathway in Fusarium graminearum and the detailed investigation into the roles of specific Tri genes. This study provides valuable insights into the complex genetic and biochemical processes involved in trichothecene production. However, to strengthen your manuscript and enhance its impact, I recommend addressing the following points:
I recommend expanding the discussion section to include a more detailed mechanistic hypothesis on how the absence of Tri3p and/or the presence of a small amount of 15-deacetylcalonectrin (15-deCAL) influences this process. This could involve exploring the potential biochemical or structural reasons behind this effect.
While your study focuses on Fusarium graminearum, it would be interesting to discuss how your findings might apply to or differ from trichothecene biosynthesis in other Fusarium/fungi species. This could help in understanding the evolutionary aspects of trichothecene biosynthesis and its regulation.
The study mentions the deletion of FgTri3 and overexpression of Tri6 and Tri10 without resulting in the accumulation of 15-deCAL. It would be beneficial to provide a more detailed analysis of the effects of Tri6 and Tri10 overexpression on the entire trichothecene biosynthetic pathway, not just the accumulation of 15-deCAL.
Implications for Agricultural Practices: Considering the importance of trichothecenes in plant pathology and agriculture, a section discussing the potential implications of your findings for controlling Fusarium infections in crops would be highly valuable. This could include strategies for genetic engineering or targeted fungicide development.
There are a few instances where technical terms and gene names might be confusing or inconsistently used.
I've noticed some challenges in the manuscript regarding the use of English that could potentially obscure the clarity and impact of your research findings. It might be beneficial to seek assistance from a professional scientific editing service or a native English speaker with expertise in your field to ensure that the language is clear, accurate, and adheres to scientific standards.
Addressing these points would significantly strengthen your manuscript, making your valuable research clearer and more accessible to readers in the field of fungal genetics and secondary metabolism.
Comments on the Quality of English LanguageI've noticed some challenges in the manuscript regarding the use of English that could potentially obscure the clarity and impact of your research findings. It might be beneficial to seek assistance from a professional scientific editing service or a native English speaker with expertise in your field to ensure that the language is clear, accurate, and adheres to scientific standards.
Reviewer 2 Report
Comments and Suggestions for Authors
Overall the manuscript is well written and an impressive amount experiments and analyses were performed. I only have a few comments:
Major comments:
· An table with an overview of the genes part of the metabolic cluster would be helpful for the reader since all the tri genes are easy to confuse.
· Line 31: I am a bit loss with this conclusion, why would overexpression of the regulators tri6 and tri10 result in the accumulation of 15-decal?
· The IDT feeding experiment is very interesting and that the shunt metabolites are being formed suggests that tri1p is involved. Would you suggest that a tri3p/tri1p double deletion mutant stop the formation of shunt metabolites and results in the accumulation of 15-decal?
· Line 245: A table with the strain, genotypes and references would be preferred.
